# Anisotropic Multiscale Modelling in SAE-AISI 1524 Gas Tungsten Arc Welded Joints

**Edison A. Bonifaz [1],\*** and **Ikumu Watanabe [2]**

1   Mechanical Engineering Department, Universidad San Francisco de Quito, Cumbayá 170901, Ecuador
2   National Institute for Materials Science, 1-2-1 Sengen, Tsukuba, Ibaraki 305-0047, Japan; WATANABE.Ikumu@nims.go.jp
\*   Correspondence: ebonifaz@usfq.edu.ec

**Abstract:** A transient non-linear multiscale finite element heat flow-mechanical model to determine micro residual stresses (type III) and micro plastic strains in SAE-AISI 1524 gas tungsten arc welded joints is developed. To include anisotropy by preferred crystallographic orientation or texture, the global domain was decomposed into small subdomains based on the concept of representative volume elements (RVEs). A three-dimensional numerical procedure was developed by using the coupling DREAM.3D-ABAQUS. The macro scale temperature gradient information as prescribed driven (load) boundary conditions was used to calculate the meso thermal cycles, and the meso scale temperature gradient information was used to calculate the micro thermal cycles needed in the subsequent mechanical analysis. Anisotropy was included by randomly entering in each grain of the RVE specimen either the maximum Young's modulus ($E_{max}$) in the stiffest direction <111>, or the minimum Young's modulus ($E_{min}$) in the least stiff direction <100>. Under this assumption, the averaging of the grain orientations over all grains in the textured polycrystal with greater number of grains ocurred, and the strength was diluted by the spread of orientations present. Higher Mises stresses evolved in the sample with coarse grain size (16 μm), which indicates that the strong dependence of residual micro stresses on grain size was reversed. The influence of the grain size on the response of the aggregates is clearly observed.

**Keywords:** GTA welding; sub-modeling; micro residual stresses; anisotropy; digital microstructure code; thermal gradients

## 1. Introduction

The properties of gas tungsten arc welded joints depend on its microstructural characteristics, tack weld distribution, and welding sequence [1]. The development of a consistent and reliable fusion weld microstructure is critical to precisely estimate the material properties. Material properties of welded parts are function of the welding processing parameters. In order to understand how these parts behave in load applications, the evolution of thermal cycles and residual stresses during component manufacturing must be investigated. In particular, the thermal stresses that evolve in the fusion and heat-affected zones are harmful to the integrity and service life of the welded part. Numerous welding problems, especially those concerning safety in the nuclear industry and repair of gas turbine components, are related to the development of residual stresses. The critical first step in addressing welding problems is to accurately calculate the transient temperature field that leads to the formation of unbalanced phases in and around the weld joint. Scarce research has been conducted to simulate the thermal cycles, thermal gradients, stress formation, and part distortion at the micro level.

One of the reasons for the little research is that the technological properties of fusion welds are created by concurrent effects of different physical phenomena that take place on different scales of length and time [2]. Length scales for processes and materials are generally classified as mesoscale (>100 μm), microscale (100 nm to 100 μm), and nanoscale

(<100 nm). The length can be either the internal dimension of a material (such as the crystalline grain size) or the external dimension of a component.

The characterization of mechanical and physical properties, on nano-, micro-, and meso scales is important for assessing other characteristics of materials such as yield strength, elastic–plastic deformation, time-dependent creep, residual stresses, fatigue, and fracture toughness. Even though differences among the physical and mechanical properties of macro scale materials and those of meso and micro scale materials are enormous, the data related to the properties of meso- and micro scale materials are lacking. Therefore, the mechanical and physical properties of micro- and meso scale materials should be studied carefully. In particular, understanding the mechanics and physics of materials at the micro scale is important because when the process size approaches the material grain size, materials should no longer be considered homogeneous. In order to predict the reliability of components in thermomechanical and load-bearing applications, the interaction of residual stresses with localized stress concentrations and crack-like defects must also be considered.

The simulation of polycrystalline superalloys with a microstructural basis is limited to only a few studies [3–8]. These analyses have been limited to two dimensional cases because of high computational requirements. In the past decade, however, the development of high-performance computers has facilitated the use of fully three-dimensional models [9–11]. The *structure-property* relationship (the link between microstructure and material macroscopic properties) is technologically interesting as it may provide valuable information for the design of enhanced materials [12–16]. In order to analyze material microstructures, it is necessary not only to generate reliable micro-morphologies with affordable computational grids, but also to describe the mechanical behavior of the elementary constituents and their interactions [9]. The problem of the generation of a suitable virtual microstructure, morphology, and mesh is particularly relevant when the analysis of a certain number of grains arranged in a 3D domain is of interest [9]. Predicting the strong dependence of flow stress and plastic strain on phase type and grain size are examples of this kind of analysis. Mechanical plastic models that incorporate grain size effects are length dependent.

Three-dimensional arrangements of grains are idealized to represent polycrystalline aggregates. Many elements per grain are used to represent nonuniform deformations within individual grains, seen as domains separated by high misorientation boundaries. In order to calculate macro and micro variables of stress and strain, the finite element mesh of the polycrystal is loaded by prescribed driven (load) boundary conditions. It better understood if the features of the material microstructure are considered in the modeling framework [9,17,18]. Grain-scale three-dimensional mechanical modeling of polycrystalline materials provides valuable information for the design of improved materials. Materials with randomly oriented grains are isotropic, whereas materials with texture are anisotropic. Cubic crystals and textured polycrystals are anisotropic with respect to strength, Young's modulus, creep, and plastic flow. When an isotropic material is elastically deformed, the Young's modulus is the material property that relates stress and strain. In metals, anisotropic elasticity behavior is prevalent in all single crystals.

Anisotropy, sometimes referred to as directionality, is a commonly observed phenomenon, whereby material properties are dependent on the direction of measurement [19]. Two sources that give rise to anisotropy are preferred crystallographic orientation or texture and alignment in the microstructure [19]. In a polycrystalline material, the directional dependence on properties is related to the processing techniques to which it has been subjected. Textured materials are frequently the result of processing techniques such as welding, wire-drawing, hot rolling, and heat treatments. As the crystal structures of metals are themselves anisotropic, some of this behavior becomes inherited by the material as a whole. Only in the case of a totally random texture are the effects of orientation completely nullified [19]. The properties of textured polycrystals can be calculated by averaging the values for their individual crystal orientations and weighted according to the frequency of

appearance in the texture [19]. The averaging of the grain orientations over all grains in the specimen is accounted for by the Taylor factor. For random textures, its value is between the Sachs and Taylor solutions of 2.24 and 3.06. The Schmid's critical resolved shear stress (CRSS) is used to identify crystallographic sliding systems that are activated under load. Therefore, the resistance of a crystal depends on both its orientation and the state of stress to which it is subjected. In a textured polycrystal, the effect will be diluted by the spread of orientations that are present [19].

The novelty of the present research is the development and application of a transient non-linear multiscale finite element model in the determination of micro residual stresses (type III) and micro plastic strains in gas tungsten arc welded joints. Studies related to optimization of the fusion welding process parameters, which are applied in the repairment of the gas turbine blades, are of great interest to engineers and researchers in the field. The present work is an effort to understand the link between solidification grains substructures with micro residual stresses and cooling rate. The development of microstructural models to determine the parameters of the fusion welding process that best preserve the monocrystalline nature of gas turbine blades during repair procedures is the main objective of this research. The polycrystalline samples were included in the finite element analysis using representative volume elements (RVEs) generated with the digital microstructure code DREAM.3D [18]. Similarly, on the sublevel scales, the evolution of the microstructure can be easily predicted with the phase field method by inputting the calculated numerical thermal results as the initial boundary conditions.

## 2. The Multiscale Finite Element Model

By solving the following governing differential equation, the temperature $T(x; y; z; t)$ at any location $(x; y; z)$ and time $(t)$ is calculated with Equation (1) documented in our previous work [2].

$$\frac{\partial}{\partial x}\left(k\frac{\partial T}{\partial x}\right) + \frac{\partial}{\partial y}\left(k\frac{\partial T}{\partial y}\right) + \frac{\partial}{\partial z}\left(k\frac{\partial T}{\partial z}\right) + \dot{Q} = \rho c_p \frac{\partial T}{\partial t} \tag{1}$$

Here, $k$ is the temperature-dependent thermal conductivity, $c_p$ is the temperature-dependent specific heat, $\rho$ is the density, $T$ is temperature, $t$ is time, and $\dot{Q}$ is the the internal heat source term. In the present work, $\dot{Q}$ is zero and latent heat was ignored. A temperature $T_o$ equal to 20 °C was used as the initial condition. Heat exchange between the top surface of the workpiece and the surroundings beyond the arc heat source involved consideration of both radiative and convective heat transfer as:

$$-k\frac{\partial T}{\partial y}\Big|_{top} + q(r) = h_t(T - T_s) + \sigma\varepsilon\left(T^4 - T_s^4\right) \tag{2}$$

Here, $T_s$ is the surrounding temperature, $h_t$ is the convection heat transfer coefficient at the top and all the other surfaces of the workpiece, $\sigma$ is the Stefan–Boltzmann constant, and $\varepsilon$ is the emissivity. Two ABAQUS [20] user subroutines FILM and DFLUX were written to account for convection heat losses and heat input distribution. Convection and radiation were considered separately by using the ABAQUS options * SFILM and * SRADIATE. Natural convection and radiation were considered in the six surfaces of the plate. The user subroutine FILM was written to account for natural and forced convection. The option *SFILM was used to call the subroutine FILM. Because of the flow of the shielding gas, the area directly beneath the nozzle of the welding gun experiences forced convection. A forced heat transfer coefficient equal to 242 W/m² K (see Table 1) was used in the finite element analysis. A moving Gaussian heat source $q(x,z,t)$ was used to represent the shape and power density distribution of the GTAW process. The heat source was applied over the top surface of the specimen during a period of time that depends on the welding speed ($v$).

$$q(x,z,t) = \frac{3Q}{\pi C^2}exp^{\{-3[(z-vt)^2+x^2]/C^2\}} \tag{3}$$

**Table 1.** Data used in the FE analysis.

| Property/Weld Parameter | Value |
|---|---|
| Thermal efficiency ($\eta$) | 0.65 |
| Forced heat transfer coefficient ($h_t$) | 242 W/m$^2$ K |
| Surface emissivity | 0.7 |
| Heat transfer coefficient ($h_t$) | 10 W/m$^2$ K |
| Heat source speed ($v$) | 4.25 mm/s |
| $T_o$ (preheating or room temperature) | 20 °C |
| Density steel ($\rho_w$) at $T_o$ | 7820 Kg/m$^3$ |
| Voltage | 10 Volts |
| Intensity | 150 Amp |
| Distribution parameter, $C$ | 4.5 mm |
| Maximum Young's modulus, $E_{max}$ <111> Stiffest direction | 273 GPa |
| Minimum Young's modulus, $E_{min}$ <100> Least stiff direction | 125 GPa |

Here, $Q = \eta V i$, where $V$ is voltage, $i$ is electric current, and $\eta$ is arc efficiency, which accounts for the radiation and other losses from the arc to the environment. The temperature-dependent material properties of the SAE-AISI 1524 carbon steel (W_S355J2G3) obtained from reference [21] are plotted in Figure 1. Other properties are summarized in Table 1. Because of its high tensile strength and lower carbon content, this steel can be easily welded.

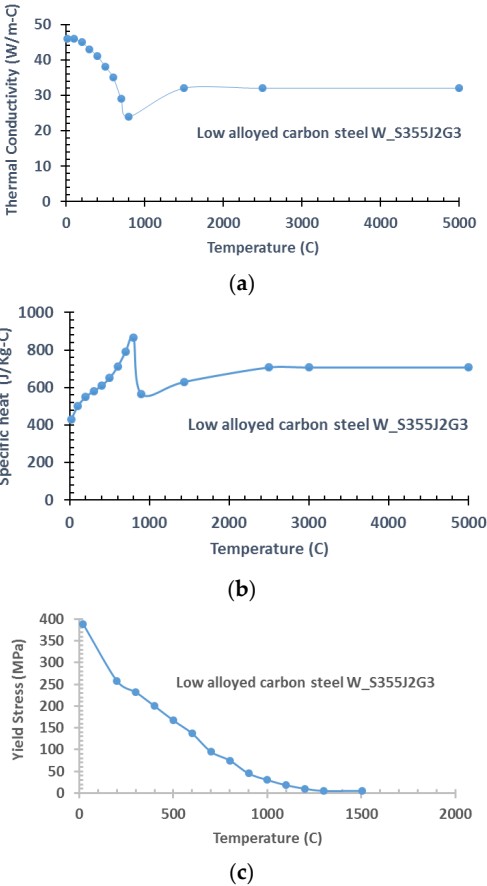

**Figure 1.** Properties of the SAE-AISI 1524 carbon steel: (**a**) thermal conductivity, (**b**) specific heat, (**c**) yield stress.

Using the welding parameters documented in Table 1, a weld was deposited in a SAE-AISI 1524 carbon steel plate. The distribution parameter *C* was selected from an experimental measurement of the fusion zone width. Cross-sectional metallographic melting temperature profiles were identified. The distribution and shape of the heat input from the heat source largely depends on its value. From there, a 3D transient-nonlinear thermomechanical analysis was carried out to determine the distribution of temperature and residual stresses during the entire welding and cooling cycle. The values of thermal efficiency, forced heat transfer coefficient, surface emissivity, and heat transfer coefficient shown in Table 1, were obtained from the literature that is documented in reference [22].

The yield condition for a multiaxial stress state (when more than one direct stress exists) is identified by the Mises stresses and equivalent plastic strains. If the material sample contains a large enough number of grains, there is no preferred crystallographic orientation, but the orientation changes randomly from one grain to the next; a reasonable physical interpretation is that the macroscale performance of the material will be isotropic. This is a further cornerstone of the Von Mises yield criterion [23]. In this manner, the single slip plasticity assumption and the von Mises yield criterion that implies plastic isotropy of the material was considered in the FE model. Anisotropy was included by randomly entering in each grain of the RVE specimen either the maximum Young's modulus ($E_{max}$) in the stiffest direction <111>, or the minimum Young's modulus ($E_{min}$) in the least stiff direction <100>.

The residual stresses and strains at any location is calculated by solving the following equilibrium governing partial differential equation:

$$\frac{\partial \sigma_{ji}}{\partial x_j} = 0 \qquad (4)$$

Here, $\sigma_{ji}$ is the stress tensor. A general framework for the mechanical analysis is provided in our previous research [22,24]. As per Reference [25], the edge size ($L_o$) of a cubic box used to represent the polycrystalline aggregate is calculated as:

$$L_o = \frac{\langle d_{gr} \rangle \left( n_{gr} \right)^{1/3}}{0.7} \qquad (5)$$

where $<d_{gr}>$ is the average grain size and $n_{gr}$ is the number of grains. In the absence of weld microstructures obtained experimentally or numerically (using phase field techniques), two polycrystalline RVEs of edge size $L_o$ = 50 μm, different grain morphology, and number of grains 11 and 44 each were randomly generated with the code DREAM.3D [18]. By using the Equation (5), the dimensions of the average grain size in the virtual specimens are 16 and 10 μm respectively. These affordable computational meshes shown on Figure 2 were incorporated in order to study the dependence of plastic strain and flow stress on grain size. The parameters to estimate the number of grains in the micro virtual specimen generated with DREAM. 3D can be operated either in the Statsgenerator or in the Initialize Synthetic Volume filter libraries. The creation of these micro virtual specimens (see Figure 2) simplifies the analysis by assuming that either of the two synthetic microstructures could result from the welding solidification process. Their origin coordinates are those located in a required position into the meso RVE specimen (see Figure 3).

A physically based multiscale model constructed to simulate the influence of the thermal cycle in the entire welding process can provide new understanding of the microstructure and distortion evolution. The multiscale frame covers thermal, micromechanical stress, and failure analysis of extended microstructural regions. In recent investigations [26,27], dendritic morphology evolution, micro residual stresses, and dislocation evolution have been integrated into a fluid-thermal-metallurgical-mechanical multiscale approach. Here, in order to mesh a local part of the model with a refined mesh based on interpolation of the solution from an initial coarse thermo macro global model (see Figure 3), the sub-modeling technique (a flexible strategy used to calculate temperature gradients, plastic strains and

residual stresses to any number of levels) was used. The meso-sub-model is the global model for the subsequent micro sub-model. With the multiscale submodeling approach, weld pool features at the macro- and meso-cale level, and micro residual stress and secondary dendrite arm spacing features at the micro scale level can be easily obtained. The prediction of the distribution and evolution of even microscopically small crystal defects such as dislocations, is another advantage of the proposed multiscale model.

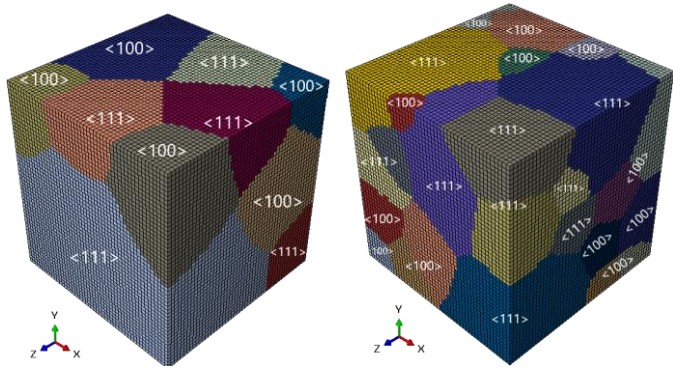

**Figure 2.** Two synthetic polycrystalline aggregates of 11 and 44 grains generated with the code DREAM.3D. The stiffest direction <111> and the least stiff direction <100> randomly assigned on each grain are also displayed.

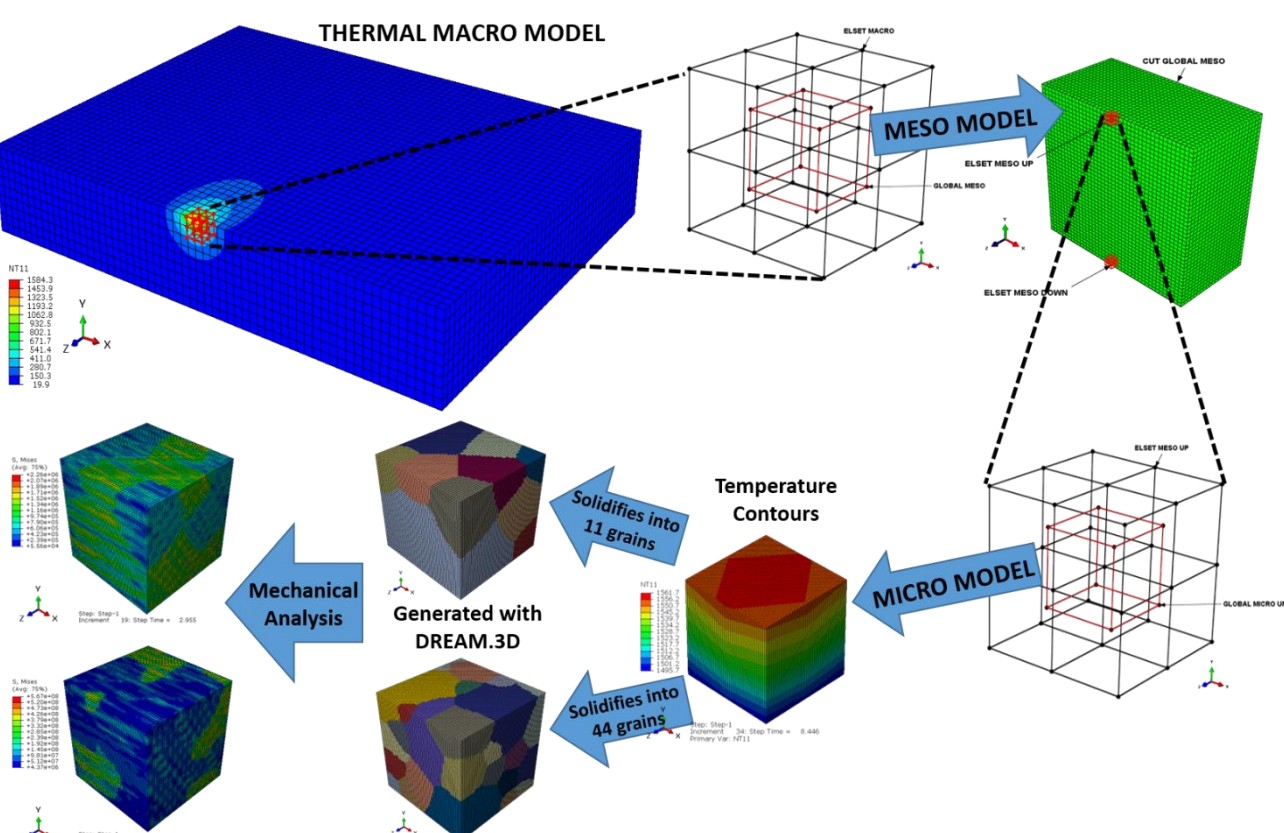

**Figure 3.** The multiscale mesh strategy applied in an initial coarse thermal macro global model. The zoomed-in volume illustrates the finite element submodeling technique. The (representative volume elements) RVE-finite element polycrystalline meshes were constructed by using the coupling DREAM.3D-ABAQUS. The coarse thermal macro mesh is composed of 48,600 DC3D8 2 mm elements. The RVE micro finite element mesh is composed of 125,000 C3D8 1 μm elements for the mechanical analysis, and 125,000 DC3D8 1 μm elements for the thermal analysis.

Material microstructures are available in many different shapes and sizes and their features of interest have different dimensionalities. Data describing attributes of microstructure can be obtained using many different devices (optical microscopy, scanning electron microscopy, electron backscatter difraction, transmission electron microscopy, wavelength dispersive spectroscopy, energy dispersive spectroscopy, atomic force microscopy, 3D Atom Probe, etc.,) [28–30]. By abstracting the materials interpretation of the features and focusing only on how the feature is described digitally, the code DREAM.3D has been able to constitute a general, unified structure for digital data that assumes no previous knowledge of material class or length-scale [18].

### 3. Results and Discussion

The technological properties of fusion welds are the result of the simultaneous effects of several physical phenomena that occur on different length scales. In the multiscaled modelling method designed to make microstructure modelling more tractable, a fine meso sub-mesh resides in each element of a coarse macroscopic global mesh. In the same manner, a fine micro sub-mesh resides in each element of a coarse mesoscopic global mesh. By performing 3D transient non-linear heat flow-mechanical simulations, thermal histories, thermal cycles, plastic strains, and stresses that evolve during the process at the sub-level scale were numerically investigated. Weld pool features at the macro and meso scale level, and micro residual stresses and plastic strains at the micro scale level were captured with the multi-scale submodeling approach. Figure 4 shows temperature distribution, micro residual stress, and equivalent plastic strain (PEEQ) distributions at documented step times in two virtual specimens of 11 and 44 grains generated with the codes DREAM.3D and ABAQUS. The dependence of micro residual stresses and equivalent plastic strains on grain size in specimens subjected to similar thermal gradient initial conditions are clearly observed. The representative micro finite element mesh is composed of 125,000 C3D8 1 μm elements for the mechanical analysis, and 125,000 DC3D8 1 μm elements for the thermal analysis.

At all considered step times, higher Mises and PEEQ values arised in the coarse grains ($d_{gr}$ = 16 μm) polycrystalline aggregate. The high values of microscale residual stresses (type III) that evolved in regions within the grains, resulted in the formation of slip bands (see Figure 4b). Intra-granular stresses are of paramount importance because of crack nucleation and propagation frequently initiate at the grain level. For these reasons, the knowledge and understanding of residual stress across the scales (Types I, II and III) is crucial for improving the accuracy of mechanical failure prediction [31,32]. The thermal gradient contours shown on Figure 4a were used as prescribed driven initial load conditions to calculate the stress and strain contours plotted on Figure 4b,c. The thermal cycles were calculated in the cooling stage of the thermal analysis. Peak cooling temperatures of 496 °C, 307 °C, 53 °C, and 33.5 °C are reported at documented step times of 9.98 s, 11.14 s, 44 s, and 120 s respectively. At these same step times, peak Mises values of 553 MPa, 644 MPa, 823 MPa and 840 MPa for the 11 grains specimen, and 495 MPa, 434 MPa, 567 MPa and 567 MPa for the 44 grains specimen are shown on Figure 4a,b respectively. Results of Figure 4 demonstrate that materials display strong size effects when the characteristic length scale associated with non-uniform local stresses and plastic strains is on the order of microns.

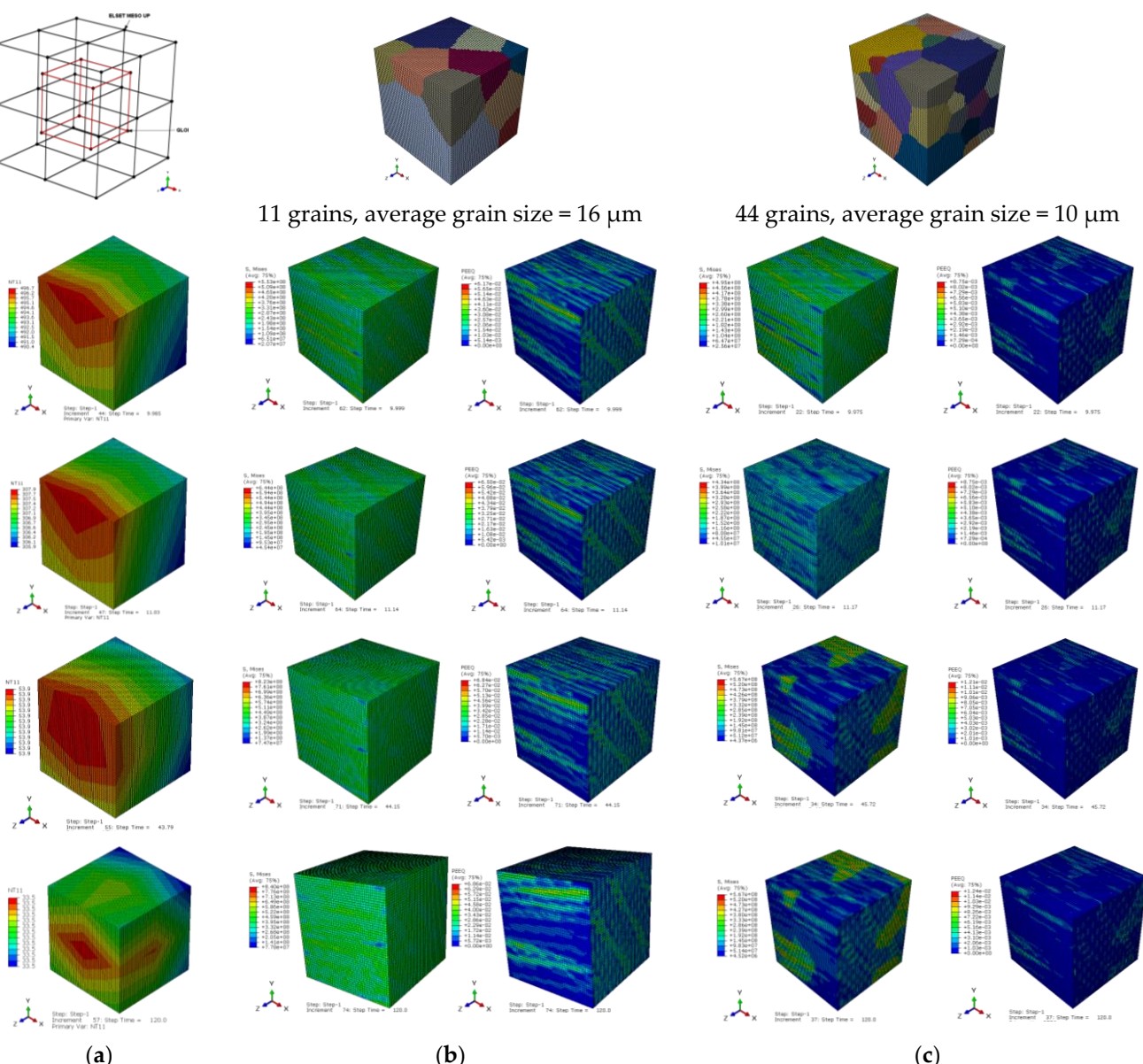

**Figure 4.** Temperature, stress, and strain distributions in representative volume elements (RVEs) at documented step times; (**a**) thermal gradient contours (**b**) micro residual stresses and equivalent plastic strains in the 11 grains aggregate; (**c**) micro residual stresses and equivalent plastic strains in the 44 grains aggregate. The units of Mises residual stresses are in Pascal.

The influence of the grain size on the aggregate response in terms of plastic strain and local stress variations are presented in Figures 5 and 6. In Figure 5b related to the 44 grain sample, it is observed that the Mises stresses decrease as the step time increases. That is, highest Mises stresses evolved in small time intervals, and as the step time increases, these stresses decreased. However, an exception to this generalized behavior occurs in the central part of the analyzed path 1 where many grains come together (see the central top surface in the 44 grains sample of Figure 2). The opposite behavior is observed in Figure 5a related with the coarse grain sample of 11 grains, where withouth any excepcion, the evolution of both, plastic deformations and Mises stresses increases as the step time increases. Higher Mises stresses evolved in the sample with coarse grain size (11 grains, $d_{gr}$ = 16 μm), which indicates that the strong dependence of residual micro stresses on grain size was reversed. The averaging of the grain orientations over all grains in the textured polycrystal with greater number of grains occurred and the strength was diluted by the spread of orientations present.

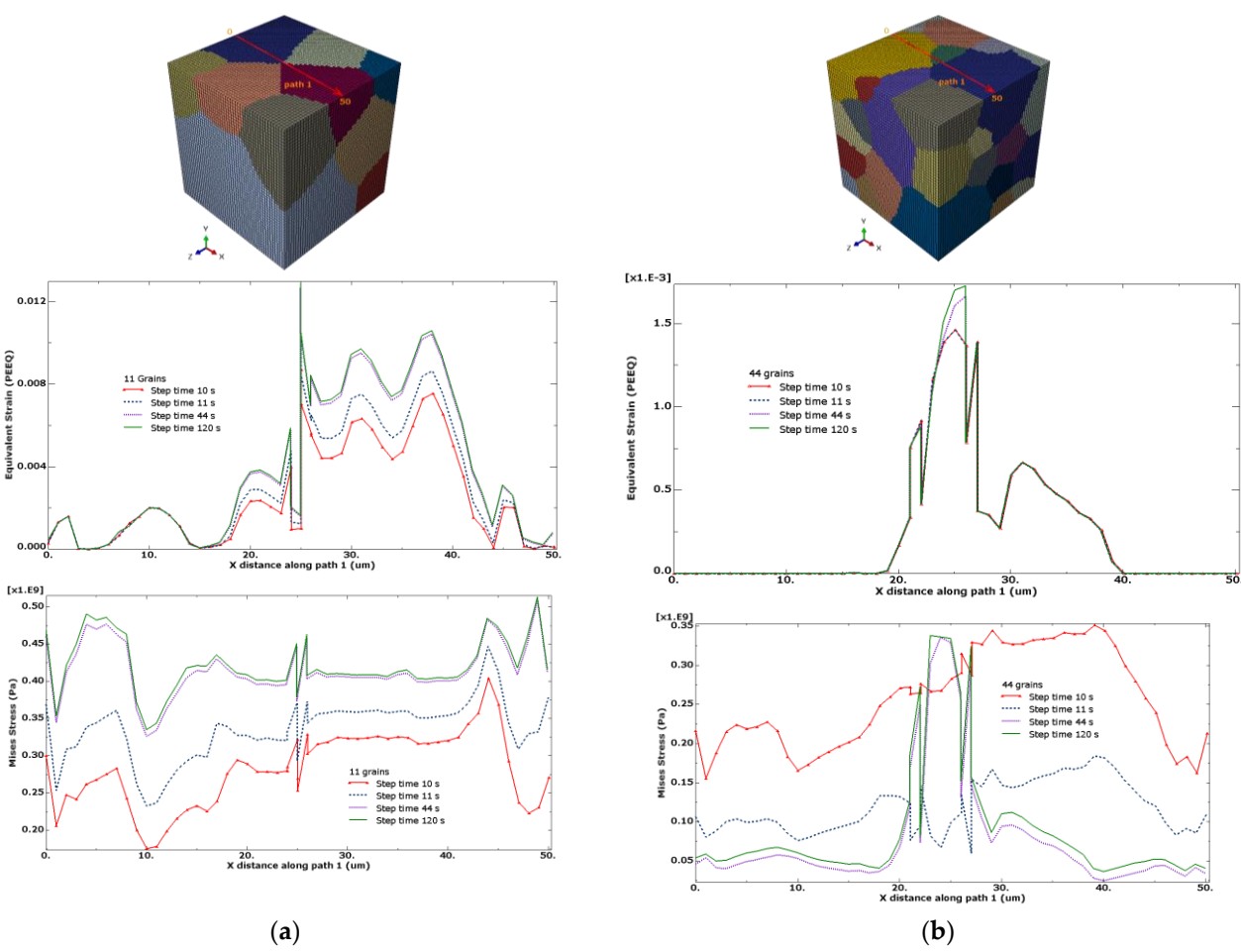

**Figure 5.** Equivalent strain (PEEQ) and Mises stress vs. X distance along path 1 at documented step times (**a**) for the 11 grains specimen (**b**) for the 44 grains specimen.

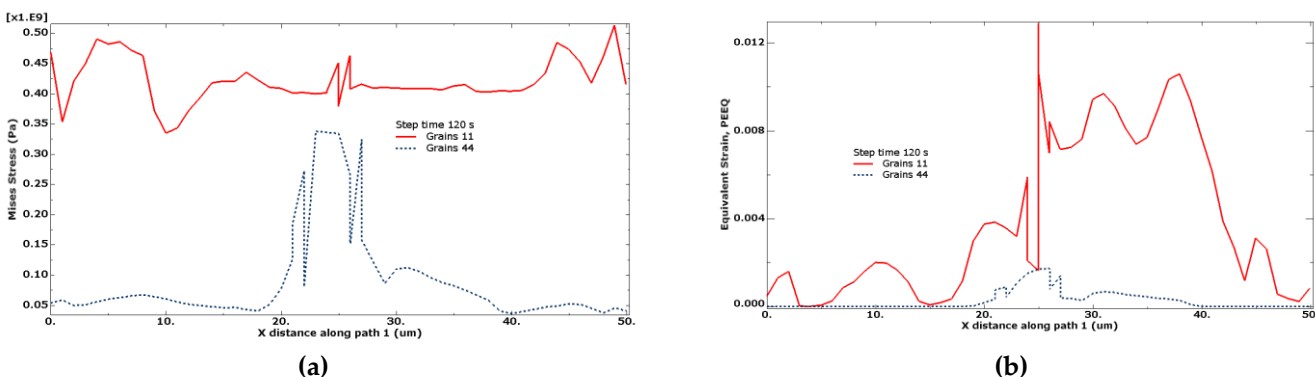

**Figure 6.** Mises stress and equivalent strain (PEEQ) vs. X distance along path 1 at the step time 120 s for the analyzed RVE specimens; (**a**) Mises stress values in Pa; (**b**) equivalent strain (PEEQ) values.

## 4. Conclusions

1. A transient non-linear multiscale finite element heat flow-mechanical model to determine micro residual stresses and corresponding plastic strains in SAE-AISI 1524 gas tungsten arc welds was developed.

2. Anisotropy was included in the FE analysis by randomly entering in each grain of the RVE specimen either the maximum Young's modulus ($E_{max}$) in the stiffest direction <111>, or the minimum Young's modulus ($E_{min}$) in the least stiff direction <100>.

Under this assumption, the averaging of the grain orientations over all grains in the textured polycrystal with greater number of grains ocurred, and the strength was diluted by the spread of orientations present.

3.  Higher Mises stresses evolved in the sample with coarse grain size (16 μm), which indicates that the strong dependence of residual micro stresses on grain size was reversed.

4.  The influence of the grain size on the response of the aggregates is clearly observed. Materials display strong size effects when the characteristic length scale associated with non-uniform local stresses and plastic strains is on the order of microns.

**Author Contributions:** Investigation, E.A.B. and I.W.; Writing—original draft, E.A.B.; Review & editing, I.W. All authors have read and agreed to the published version of the manuscript.

**Funding:** This research received no external funding.

**Data Availability Statement:** Not applicable.

**Conflicts of Interest:** The authors declare no conflict of interest.

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
