# Peer review of "Anisotropic Multiscale Modelling in SAE-AISI 1524 Gas Tungsten Arc Welded Joints"

_crystals, doi:10.3390/cryst11030245_

Round 1

Reviewer 1 Report

The authors adopted all suggestions and the manuscript has been significantly improved. The submitted manuscript is worth publishing in Crystals.

Author Response

No comments, no response.

Reviewer 2 Report

The paper is of great interest to the reader as employs a multiscale model framework to understand and simulate residual stress evolution during a thermal cycle.

Nevertheless, the paper the reviewer cannot suggest acceptance of the paper in this form since it lacks any experimental evidence that the results are sound. On top of that, the results do not appear very appealing in the way they are reported. For these reasons, I cannot suggest acceptance of this paper in this form. A deep revision is required.

The major concerns requiring careful attention are:

  1. It is extremely important to classify residual stress across the lengthscales. Given the scale, residual stress is classified into three types, i.e. Type I,II and III [1, 2]. This should be mentioned in the introduction in the context of the aim of this paper.
  2. For instance, Figure 4 does not show any useful information for two reasons: 1) The font size is extremely small, and the reader cannot understand; 2) Contour plot of Von Mises stress does not seem the best choice, perhaps. It would be more appropriate to show the principals stresses (at least the first principal) and show the principal directions. In this way it would be possible to better understands the origin of residual stress
  3. Another results that is very important to show is the statistical distribution of residual stress at the integration points (or at least in the grains) to see whether it resembles the experimental findings. Please read and refer to the following papers: [2-4]

References:

[1] L. Pintschovius, V. Jung, E. Macherauch, O. Vöhringer, Residual stress measurements by means of neutron diffraction, Materials Science and Engineering 61(1) (1983) 43-50.

[2] E. Salvati, A.M. Korsunsky, An analysis of macro- and micro-scale residual stresses of Type I, II and III using FIB-DIC micro-ring-core milling and crystal plasticity FE modelling, International Journal of Plasticity 98 (2017) 123-138.

[3] H. Zhang, A. Jérusalem, E. Salvati, C. Papadaki, K.S. Fong, X. Song, A.M. Korsunsky, Multi-scale mechanisms of twinning-detwinning in magnesium alloy AZ31B simulated by crystal plasticity modeling and validated via in situ synchrotron XRD and in situ SEM-EBSD, International Journal of Plasticity 119 (2019) 43-56.

[4] J. Chen, A.M. Korsunsky, Why is local stress statistics normal, and strain lognormal?, Materials & Design 198 (2021) 109319.

Author Response

1) The words: Type III and micro,  were added to the text. 

2) The following paragraph was added to the text to better understand the meaning of the Fig. 4:

At the same step times, peak Mises values of 553 MPa, 644 MPa, 823 MPa and 840 MPa for the 11 grains specimen; and 495 MPa, 434 MPa, 567 MPa and 567 MPa for the 44 grains specimen are shown on Fig. 4a and Fig. 4b respectively.

3) N/A in this research.

Round 2

Reviewer 2 Report

The authors have not addressed my concerns properly, none of them. The response letter appears very superficial. I would like to give them a one more chance to improve their paper. If the response will be the same, I will recommend rejection.

1- the authors now refer to type III Residual stress but they do not provide any reference to papers that describe how this classification is obtained. Please refer to the papers I pointed out in my previous review.

2- My concern about Fig.4 is still on. The authors have not done anything to improve it. In the current for it simply does not help the reader to understand. 

3- the authors just reply to my 3rd point by saying that it is not applicable. They should at least explain why.

Author Response

1) The suggested papers were added to the references section (See references 31 and 32 in red color).

2) The following paragraph was added to explain much better what is observed in Fig. 4:

The high values of microscale residual stresses (type III) that evolved in regions within the grains, resulted in the formation of slip bands (see Fig. 4b). Intra-granular stresses are of paramount importance because of crack nucleation and propagation frequently initiate at the grain level. For these reasons, the knowledge and understanding of residual stresses across the scales (Types I, II, and III) is crucial for improving the accuracy of mechanical failure prediction [31, 32]

3) Statistical and Experimental validation will be performed in a future research work. The focus of the present research was the development and application of a transient non-linear multiscale finite element model in the determination of micro residual stresses and micro plastic strains in gas tungsten arc welded joints.

This manuscript is a resubmission of an earlier submission. The following is a list of the peer review reports and author responses from that submission.

Round 1

Reviewer 1 Report

Dear Author,

I have reviewed paper titled "Anisotropic Multiscale Modelling in SAE-AISI 1524 gas tungsten arc welds", which was submitted to Crystals.

The paper presents some valuable results. After required improvements it could be considered for publishing. However, in this state the paper should be rejected. I have some suggestions, which are listed below. 

General remarks:

  • Please check your references. I can find different styles, wide from template of the journal.
  • You have presented 21 references. Most of them are old articles, only a few has been published ater 2018. You should extended this list to underline the new published information. The science made big step forward last years, which should be marked in your paper.
  • I propose change the title. Welds meen only one area of welded joint (weld metal). You have investigated whole joint, so the title should be "Anisotropic Multiscale Modelling in SAE-AISI 1524 gas tungsten arc welded joints".
  • Delete interlayer between lines 29-30.
  • In keywords I propose add GTA welding or TIG welding.

Introduction:

  • Lines 36-39 - the same info has been presented many times.
  • For the properties of TIG welded joints, their residual stresses and distortion some more factors are responsible, not only microstructural characteristic, as you stated - e.g.m tack weld distribution and welding sequence (http://casopisi.junis.ni.ac.rs/index.php/FUMechEng/article/view/6362).
  • Please clearly mark the novelty of your work in this section. Now, it is hard to understand, what new has been proposed.

The multi-scale finite element model:

  • Please add reference to presented equations.

Results and Discussion:

  • It is not clear, why 11 and 44 grains were taken for your research. Please support with relevant statements.
  • Fig. 4 - it is imposible to see the values from the pictures. Please rebulit this figure.
  • I cannot see any scientific discussion here. You did not compare your results and their advantages compared to other scientiests.
  • The presented range of results is very poor. The paper looks like technical report.

Conclusions:

  • It should be extended.

Author Response

General Remarks

1) References were checked and modified according (see lines 302 to 346)

2) Six new published articles were included in the manuscript. 

3) The title was changed with the words welded joints.

4) GTA welding was included in the keywords.

Introduction

1) The first sentence of the first paragraph in the Introduction section was modified. 

2) The novelty of the work is explained in lines 108-121

The multiscale finite element model

1) A reference was added.

Results and discussions

Lines 185, 189-191 explain why 11 and 44 grains were taken in the research.

The results are improved with a scientific discussion stated in lines 233-238 and 255 to 261.

The quality of Fig. 4 is OK. With a zoom, the values can be seen. 

The conclusions in the present form satisfied the opinion of the other two reviewers.

Reviewer 2 Report

Manuscript titled "Anisotropic Multiscale Modelling in SAE-AISI 1524 gas tungsten arc welds" presents the development and application of transient non-linear multiscale finite element model in the determination of micro residual stresses and corresponding plastic strains in SAE-AISI 1524 carbon steel gas tungsten arc welds. Studies related to optimization of the fusion welding process parameters, which are applied in the repairment of the gas turbine blades, are of great interest to engineers and researchers in the field. Hence, the submitted manuscript is worth publishing in Crystals.   To improve the impact of the presented study, it would be good to revise the current version of the manuscript according to the following suggestions: 1. Abstract should be revised according to Instruction for Authors (https://www.mdpi.com/journal/crystals/instructions#preparation). The abstract should be a total of about 200 words maximum. In current version is more than 300 words. 2. All abbreviations used in the text should be explained such as FE (finite element) etc. Further, um should be written with the Greek letter µ as µm. Also, multi-scale should be written as multiscale uniformly throughout the text. 3. In section 1 (Introduction - Line 69) it is written: "...to only a few studies [2]." Here, the author should provide more references than one, reference B. Adams et al. (1998). 4. The last paragraph in section 2 (Lines 202-208) should be shifted at the end of section 1 (Introduction). 5. The Acknowledgement part is missing.

Author Response

1) The abstract was reduced.

2) All abbreviations were explained. 

3) um was replaced with μm.

4) multi-scale was replaced with multiscale.

5) Five new references were added in the phrase "...to only a few studies [3-8]

6) The last paragraph in section 2 was shifted at the end of section 1

7) No acknowledgement part in this research.

Reviewer 3 Report

Comment 1: Regarding Equation (2),

Equation (2) simply describes convective heat transfer and radiative heat transfer. In general, welding heat transfer analysis reflects convection and radiative heat transfer in two ways. The author is thought to have used (2) out of the following two methods.
(1) Convection and radiation are replaced with equivalent convective heat transfer coefficients.
(2) Convection and radiation are considered separately.

If Author has used (2), please explain the area to which the convective boundary condition and the radiative boundary condition are assigned separately. Also, please explain in more detail how to analyze radiative heat transfer in FEA.

Comment 2: Regarding Table 1. 

The reviewer believes the following values are the most important values in numerical welding mechanics. Please explain in detail the process and basis for determining the characteristic value.

Thermal efficiency, Forced Heat transfer coefficient, Surface emissivity. Heat transfer coefficient, Distribution parameter in Table 1.

Author Response

1) Convection and radiation are considered separately. Lines 136 to 143 were added to explain the reviewer comments.

2) The values documented in Table 1 are explained in lines 155 to 162.

Round 2

Reviewer 1 Report

Dear Author,

I heve read the revised version of your paper. In my opinion the paper has been improved a lot. The most important, the scientific discussion has been extended. Also, the novelry is clearly stated in this version of the manuscript.

In my opinion, the Fig. 4 should be bigger. However, I agree with you that with a zoom, the values can be seen. Following this, it can be pubished in this state.

Congratulations for really good improvements.
